# *Thymus vulgaris* L. Essential Oil Solid Formulation: Chemical Profile and Spasmolytic and Antimicrobial Effects

**DOI:** 10.3390/biom10060860

**Published:** 2020-06-04

**Authors:** Matteo Micucci, Michele Protti, Rita Aldini, Maria Frosini, Ivan Corazza, Carla Marzetti, Laura Beatrice Mattioli, Gabriella Tocci, Alberto Chiarini, Laura Mercolini, Roberta Budriesi

**Affiliations:** 1Department of Pharmacy and Biotechnology, Nutraceutical Lab, Alma Mater Studiorum—University of Bologna, 40126 Bologna, Italy; matteo.micucci2@unibo.it (M.M.); rita.aldini@unibo.it (R.A.); laubea91@gmail.com (L.B.M.); alberto.chiarini@unibo.it (A.C.); 2Department of Pharmacy and Biotechnology, Research Group of Pharmaco-Toxicological Analysis (PTA Lab), Alma Mater Studiorum—University of Bologna, 40126 Bologna, Italy; michele.protti2@unibo.it; 3Department of Life Sciences, Vita, University of Siena, 53100 Siena, Italy; maria.frosini@unisi.it; 4Department of Experimental, Diagnostic and Specialty Medicine—DIMES, Alma Mater Studiorum—University of Bologna, 40138 Bologna, Italy; ivan.corazza@unibo.it; 5Valsambro S.r.l., Via Cairoli 2, 40121 Bologna, Italy; carla.marzetti@valsambro.it (C.M.); gabriella.tocci@valsambro.it (G.T.); 6GVM Care & Research, 48022 Lugo, Ravenna, Italy

**Keywords:** capillary electrochromatography, diarrhoea, intestinal contractility, L-type Calcium channels, LC-MS/MS, solid based formulation

## Abstract

A new *Thymus vulgaris* L. solid essential oil (SEO) formulation composed of liquid EO linked to solid excipients has been chemically analysed and evaluated for its intestinal spasmolytic and antispastic effects in ex vivo ileum and colon of guinea pig and compared with liquid EO and excipients. Liquid EO and solid linked EO were analysed by original capillary electrochromatography coupled to diode array detection (CEC-DAD) and liquid chromatography-tandem mass spectrometry (LC-MS/MS) methodologies. The main bioactive constituents are thymol and carvacrol, with minor constituents for a total of 12 selected analysed compounds. Liquid EO was the most effective in decreasing basal contractility in ileum and colon; excipients addiction permitted normal contractility pattern in solid linked EO SEO. In ileum and colon, the *Thymus vulgaris* L. solid formulation exerted the relaxant activity on K^+^-depolarized intestinal smooth muscle as well as liquid EO. The solid essential oil exhibits antimicrobial activity against different strains (*Staphylococcus aureus, Streptococcus pyogenes*, *Pseudomonas aeruginosa*, *Escherichia coli*, *Salmonella Thyphimurium*, *Candida albicans*) similarly to liquid oil, with activity against pathogen, but not commensal strains (*Bifidobacterium Breve*, *Lactobacillus Fermentum*) in intestinal homeostasis. Therefore, *Thymus vulgaris* L. solid essential oil formulation can be proposed as a possible spasmolytic and antispastic tool in medicine.

## 1. Introduction

In traditional medicine, humans have greatly benefited from plants and their secondary metabolites. Plants have been used not only for their properties occasionally attributed to their secondary metabolites, but interest is also due to essential oils (EOs) diffused in or obtained from the surface of plant organs, particularly their aerial parts—flowers and leaves. EOs are complex mixtures of chemical compounds that have long been known and used as natural food preservatives, aromatic additives [1], in personal care products, and in aromatherapy [2]. The interest of the scientific community in EOs has been increasing, as their therapeutic properties have been gradually confirmed [1]. Among all the EOs used in the therapeutic field, there is a growing commercial interest in the EO of *Thymus vulgaris* L. (common thyme), now one of the 10 most widely traded EOs in the world. It is an aromatic and medicinal plant. So far, 928 species of the *Thymus* genus have been identified in Europe, North Africa, Asia, South America, and Australia [3]. *Thymus* EOs are mainly composed of terpenes, terpene alcohols, esters, and phenolic derivatives. Among them, thymol and carvacrol present the highest interest. The traditional and the therapeutic use, supported by in vitro studies, has shown that *Thymus* and its derivatives present a wide biological spectrum—antioxidant [4,5], antibacterial [5,6], antifungal [7], and muscle antispasmodic activities [8].

*Thymus* EOs have beneficial effects on neurodegenerative, cardiovascular, cancer, and inflammatory diseases [9]; are traditionally used to treat respiratory tract problems [8]; and relieve gastrointestinal spasm and digestion [3]. Therefore, *Thymus* and its EO are functional and promising in medicine [3].

In this paper, we have studied a peculiar formulation of *Thymus vulgaris* L. EO absorbed onto a solid matrix of excipients (solid essential oil—SEO) inserted into a capsule (operculum) for a potential use in intestinal pathologies. The operculum was opened, and the solid liquid oil was obtained. The rationale was to obtain information about the effect of this solid form of EO on intestinal contractility. EO solid formulation, in fact, possessed many advantages, such as the ability to overcome patient taste and smell distaste; to modulate EO release; to increase the stability of the active essential oil; and to reduce volatility, toxicity, and interactions with the intestinal substances, thus improving patient compliance and convenience. In addition, an original analytical methodology based on capillary electrochromatography coupled to diode array detection (CEC-DAD) was developed for the separation and determination of twelve selected constituents of *Thymus vulgaris* L. EO as a representative set of bioactive compounds that can be correlated with the biological activities object of this study. The target analytes were thymol, carvacrol, p-cymene, α-terpinene, γ-terpinene, linalool, borneol, β-cariophyllene, β-myrcene, α-terpineol, β-pinene, and limonene. After method validation with good results in terms of linearity, precision and accuracy, this original strategy was applied for the analysis of *Thymus vulgaris* L. EO and a derived formulation. In addition, thymol and carvacrol being among the most abundant and characteristic bioactive constituents of *Thymus* EOs, an original method based on liquid chromatography-tandem mass spectrometry (LC-MS/MS) was set up and validated in order to accurately confirm thymol and carvacrol levels in the analysed samples and to demonstrate the effectiveness of CEC-DAD analysis. Since the current focus on natural products is to develop their formulation to improve bioavailability, pharmacokinetics, and to reduce adverse effects for the treatment of various human diseases [10], the main components of the formulation (SEO, EO, and excipients) were then evaluated for a potential use in intestinal pathologies, by assessing their effects on spontaneous and induced contractility of guinea pig smooth muscle isolated gallbladder, gastric fundus, ileum, and colon. Otilonium bromide (OB), an antispasmodic drug, has been considered as a positive control. At the same time, potential antimicrobial activity of *Thymus* EO alone, SEO, and excipients has been tested against some bacteria and fungi most commonly involved in the onset and progression of gastrointestinal diseases.

## 2. Materials and Methods

The *Thymus vulgaris* L. solid essential oil (SEO) formulation named Aromatoil^®^ (manifactured by Coima, Bastia (RA), Italy) used in this study was supplied by BIO-LOGICA, Via della Zecca 1, 40100, Bologna, Italy. The used essential oil has been obtained by steam distillation of the summit flowers. The formulation was made by *Thymus vulgaris* L. EO absorbed to a solid matrix of excipients (SEO) inserted into a capsule (operculum). The operculum was discharged, and the solid liquid oil SEO was obtained. Each operculum, contains summit flowers *Thymus vulgaris* L. essential oil (0.6 mg) and 340.4 mg of excipients consisting of pregelatinized corn starch, soy lecithin, ascorbic acid, calcium carbonate, levilite, vegetable magnesium stearate (Invention Patent Application N: 102018000007395—(I0174439)).

### 2.1. Instrumental Analysis: CEC-DAD and LC-MS/MS

Chemicals and solutions. Analytical-grade standards of thymol [5-methyl-2-(propan-2-yl)phenol], carvacrol [2-methyl-(propan-2-yl)phenol], p-cymene [1-methyl-4-(propan-2-yl)benzene], α-terpinene [4-methyl-1-(1-methylethyl)-1,3-cyclohexadiene], γ-terpinene [4-methyl-1-(1-methylethyl)-1,4-cyclohexadiene], linalool (3,7-dimethyl-1,6-octadien-3-ol), borneol (endo-1,7,7-trimethyl- bicycle[2.2.1]heptan-2-ol), β-cariophyllene {(1R,4E,9S)-4,11,11-Trimethyl-8-methylidenebicyclo[7.2.0]undec-4-ene}, β-myrcene (7-methyl-3-methylene-octa-1,6-diene), α-terpineol [2-(4-Methylcyclohex-3-en-1-yl)propan-2-ol], β-pinene (6,6-dimethyl-2-methylidenebicyclo[3.1.1]heptane), limonene [1-methyl-4-(prop-1-en-2-yl)cyclohex-1-ene] and xylene (dimethylbenzene), used as the internal standard (IS) for CEC-DAD, were obtained from Sigma Aldrich (St. Louis, MO, USA). MS-grade acetonitrile (ACN) and methanol (MeOH), analytical-grade formic acid, and ammonium acetate were also purchased from Sigma Aldrich, while ultrapure water (18.2 MΩ·cm) was obtained by means of a Milli-Q system from Millipore (Burlington, MA, USA). Analyte and IS stock solutions (1 mg/mL) were prepared in MeOH and stored at −20 °C, working solutions were prepared daily by dilution in the mobile phase of each system and stored protected from light in amber glass vials.

#### 2.1.1. Analytical Conditions

CEC-DAD analyses were carried out on a ^3D^CE capillary electrophoresis apparatus from Agilent Technologies, equipped with a DAD operating at 210 nm. Fused silica capillary (32 cm total length x 100 μm ID, 375 OD) was from Polymicro Technologies (Phoenix, AZ, US) and packed with LiChrospher 100 RP-18 endcapped particles (5 µm particle size, 100 Å pore size) from Merck-Millipore. The optimised mobile phase was a mixture of 50 mM, pH 5.5 ammonium acetate solution and ACN (15/85, V/V), while the capillary temperature was kept constant at 25 °C. Analyses were carried out applying a 30 kV voltage and 8 bar pressure at both ends of the capillary and samples were injected at the anodic end of the capillary by applying a pressure of 5 bar for 30 s. LC-MS/MS analysis was exploited in order to confirm quali-quantitative results obtained by CEC-DAD as regards major components thymol and carvacrol. The LC-MS/MS analytical conditions developed ad-hoc for this research work are described in Appendix A.

#### 2.1.2. Sample Analysis

In order to be applied for the analysis of EO and formulations, both CEC-DAD and LC-MS/MS methods were fully validated on analyte standard solutions, according to the main international guidelines [11] in terms of linearity (including limit of detection, LOD and limit of quantitation, LOQ), precision and accuracy. As regards EO analysis, a 100-μL aliquot was suitably diluted in the mobile phase of both the instrumental systems, filtered through 0.2 µm pore diameter nylon syringe filters and injected in the two analytical systems described above. As regards formulations containing Thymus EO, three capsules were individually weighed, their content was mixed, and an aliquot of 100 mg was extracted with 10 mL of mobile phase by vortex agitation for 30 s and ultrasonic bath for 15 min. The suspension was then centrifuged at 4500 RPM for 10 min, the supernatant was transferred in autosampler vials and analysed by both CEC-DAD and LC-MS/MS. Compound quantitation was obtained by integrating peak areas obtained from sample analysis and interpolation on the linearity curve of each analyte. All analyses were carried out in triplicate by both CEC-DAD and LC-MS/MS on a single batch of both EO and formulations. Quantitative results were then expressed as μg of analyte for 100 μL of EO (% m/V) for *Tymus vulgaris* L. EO samples and as µg/cps for EO-based formulations.

### 2.2. Ex Vivo Muscle Contractibility Evaluations

Male Guinea-pig (200–400 g) obtained from Charles River (Calco, Como, Italy) were used. The animals were housed according to the ECC Council Directive regarding the protection of animals used for experimental and other scientific purposes. The work was conducted according to the guidelines set forth to EU Directive 2010/63/EU and to ARRIVE guidelines [12]. The protocol was approved by the Institutional Ethics Committee of the University of Bologna (Protocol PR 21.79.14) and transmitted to the Ministry of Health. Humane end points were followed (https://www.humane-endpoints.info/en).

The animals were sacrificed by cervical dislocation. The organs studied were stomach, ileum, proximal colon and gallbladder. Briefly, the organs were set up rapidly under a suitable resting tension in 15 mL organ bath, containing appropriate physiological salt solution (PSS) consistently warmed and buffered to pH 7.4 by saturation with 95% O_2_—5% CO_2_ gas and used as previously described [13].

For detailed information about gastric fundus, ileum, proximal colon, and gallbladder, see Appendix A.

#### 2.2.1. Contractility Spontaneous

The tracing graphs of Spontaneous Contractions (SC) (g/min) of ileum, colon, gallbladder and gastric fundus were continuously recorded with the LabChart Software (version 5.04, GraphPad Software Inc., San Diego, CA, USA). After the equilibration period (about 30 min to 45 min according to each tissue) cumulative-concentration curves (0.1, 0.5, 1, 5, 10 mg/mL) to samples were constructed. At the end of each single dose, a 5 min stationary period was selected and, for each interval, the following parameters were evaluated: mean contraction amplitude (MCA), calculated as the mean force value (g); the force contractions standard deviations, considered as an index of the spontaneous contraction variability (SCV); and basal spontaneous contraction activity (BSCA), calculated as the percentage (%) variation of each mean force value (g) with respect the control. For details about spontaneous contractions rates through a standard FFT analysis, see Appendix A.

In order to avoid errors due to the presence of artefacts, the period of analysis was chosen by a skilled operator.

#### 2.2.2. Contractility

The spasmolytic activity via action on L-type calcium channel was studied using ileum, colon, and gallbladder contracted by high K^+^-concentration. Tension changes in smooth muscle relaxation were recorded isometrically as previously described [14].

### 2.3. Antibacterial Activity

The antibacterial activity was performed against Gram^+^: Staphylococcus aureus (ATCC 25923 KS2), Streptococcus pyogenes (ATCC 19615), Bifidobacterium Breve (ATCC 15700), Lactobacillus Fermentum (ATCC 9338); Gram: Pseudomonas aeruginosa (ATCC 27853), Escherichia coli (ATCC 700728), Salmonella Thyphimurium (ATCC 14028); and fungus: Candida albicans (ATCC 14053). For detailed information, see Appendix A.

### 2.4. Statistical Analysis

For in vitro studies on isolated organs, data are presented as described below. Spontaneous contractility: the samples were added in a cumulative manner. Variation higher than 10% percent variations of each range were considered statistically significant. On spontaneous contractility experiments, data from concentration-response curves were analysed by GraphPad Prism^®^ version 5.04, GraphPad Software Inc., San Diego, CA, USA [15,16]. Induced contractility: the spasmolytic activity of samples was expressed as the percent inhibition of calcium-induced contraction on K^+^-depolarized ileum, colon and gallbladder strips (smooth muscle activity) and presented as mean ± S.E.M. The potency of all samples defined as IC_50_ was evaluated from log concentration–response curves (Probit analysis by Litchfield and Wilcoxon, n = 6–8) in the appropriate pharmacological preparations [15,16,17]. Antibacterial activity: the minimal inhibitory concentrations (MICs) values were determined by the microdilution method [18]. Data were evaluated using the IBM SPSS software program (version 19; IBM SPSS Inc., IL, USA). All tested samples and control groups were compared at the 95% confidence interval.

## 3. Results

### 3.1. Analytical Characterization

In order to effectively analyse the content of representative compounds in Thymus EO and EO-based formulations, original CEC-DAD and LC-MS/MS methodologies were optimised and fully validated. CEC-DAD was exploited to perform a qualitative and quantitative evaluation of 12 compounds, while LC-MS/MS was used to accurately confirm the quantitative levels of thymol and carvacrol in samples.

Both CEC-DAD and LC-MS/MS methods were fully validated in terms of linearity, precision and accuracy. Method development and complete validation data are reported in Appendix A. Briefly, method sensitivity was between 2 µg/mL and 5 µg/mL in terms of limit of quantitation (LOQ) while linearity was deemed good (r^2^ ≥ 0.9991) over the 5–200 µg/mL range for all the analytes. Method precision was also satisfactory, being the percentage relative standard deviation (RSD%) always < 5.7%, while accuracy was ≥ 85%.

Qualitative and quantitative results obtained from the analysis of *Thymus* EOs and derived formulation, by applying CEC-DAD methodologies are reported in Table 1.

As can be seen, all 12 compounds included in the CEC-DAD method were successfully identified and quantified in both samples. As regards confirmatory analysis performed by LC-MS/MS for thymol and carvacrol, these provided quantitative results in good agreement with those obtained by CEC-DAD, namely 43.5 ± 0.3 µg/100 µL and 21.0 ± 1.1 µg/100 µL for thymol and carvacrol in EO samples respectively, and 213.3 ± 3.4 μg/cps and 102.2 ± 4.8 respectively for thymol and carvacrol in EO-based formulations, thus proving the effectiveness of CEC-DAD analysis. The electrochromatogram obtained from the analysis of a *Thymus vulgaris* L. essential oil sample under the optimised conditions is shown in Figure 1, while the LC-MS/MS chromatogram obtained from the analysis of a *Thymus vulgaris* L. essential oil–based formulation sample is shown in Figure 2.

### 3.2. Ex Vivo Muscle Contractibility Evaluations

Liquid EO and SEO effects were studied on gastric fundus, ileum, colon, and gallbladder contractility, on both spontaneous and induced contractility (K^+^ 80 mM). In addition, the excipients (Table 1) were also tested. The results were compared with antispasmodic and spasmolytic activity of OB, taken as a positive reference drug [19]. A quantitative comparison between EO, SEO, and OB is not possible, since OB is a single molecule, EO and SEO are a mixture of chemical compounds. Only a qualitative comparison is possible, most important for a possible therapeutic use.

#### 3.2.1. Spontaneous Contractility

The variation of spontaneous contraction for isolated stomach, ileum, colon, and gallbladder tissues was evaluated through concentration-response curves. For all tissues, the changes of basal activity induced by these chemical compounds were evaluated and were expressed as BSMA, SC and following the modification of frequency bands of interest in each tissue. For all tissues, the original record tract was shown.

##### Ileum

Liquid EO has the highest effect and induces a decrease (by 40%) in ileal tone as early as at 0.1 mg/mL concentration, up to a 44% at 10 mg/mL; SEO presents an effect concentration dependent, with maximal activity of 48% at 10 mg/mL. Excipients have minimal effect: up to 15% at 10mg/mL (Figure 3). At the maximal concentration, (10 mg/mL) EO and SEO effects are similar, although the EO effective concentration is 500 times lower in SEO than in EO. The OB IC_50_ (5 × 10^−7^ M) on basal spontaneous contraction activity (BSCA) is close to the effects elicited by EO and SEO at the maximal concentration.

Spontaneous contraction greatly decreases for EO at the minimal concentration, it decreases for SEO at the maximal concentration (10mg/mL), consistently for the lower dose of essential oil in SEO; spontaneous contraction decreases slightly and minimally for excipients and OB, respectively. Therefore, the excipients effect is compliant with essential oil.

Contractility decreases at the lowest concentration (0.1 mg/mL) for EO, while it remains almost constant for SEO and drops with the maximum concentration (10 mg/mL), and variability decreases at 10 mg/mL. MCA and variability progressively decrease up to the maximal concentration; mean amplitude and variability progressively decrease for OB (Figure 4).

Therefore, liquid essential oil has the highest power of decreasing the ileal contractility, since the solid formulation presents 0.18% of essential oil rather than the liquid form. Excipients do not get in the way of essential oil, but they act synergistically. Moreover, spontaneous contraction rates (FFT) analysis showed that variability drops from the control to the first concentration and then remains unchanged (on all frequencies) with a minimum for 10 mM concentrations (Appendix A). OB maintains the ileal contraction, but in presence of a modest decrease of contraction amplitude and low frequencies waves. Therefore, SEO profile seems similar to OB profile, since the decrease in ileal tone seems associated to modest reduction in waves morphology (Appendix A).

##### Colon

Liquid EO reduces by 44% the ileal tone, independently on the dose; SEO decreases the tone in a concentration dependent manner up to 50% at the highest dose. Excipients effect is much less significant, as it reached the maximum of 20% at the highest concentration of 10 mg/mL. OB decreases the tone, dose dependently up to 30% at 5 × 10^−5^ M (Figure 5).

Spontaneous contraction is decreased severely by EO, less by SEO, even less by excipients, and not by OB. Mean contraction amplitude is severely reduced by EO, with minimal variability; it is progressively reduced by SEO, with reduced variability; and by excipients, reduced by OB only at the highest concentration but without variability (Figure 6). In addition, the size of the bars in the FFT are very high for the control and smaller but similar to each other for the different concentrations. (Appendix A).

Excluding the control (for which the variability is very different), the contractility for SEO and OB is similar.

The effect of SEO is similar in the two organs, liquid EO is the most effective in both the organs; excipients decrease the effect of EO to the values of SEO. The SEO decrease in intestinal tone is 50% both in the ileum and in the colon and is similar to OB in ileum.

Ileal and colonic contraction is maintained in presence of a reduction of the tone, although the SCV is maintained more in the ileum than in colon, suggestive of a stronger effect on the colon.

##### Gallbladder

Liquid EO and SEO decrease the BSCA independently on concentration at minimal value (about 5%). Surprisingly, a dose-dependent effect is obtained by excipients, but without influence on SEO values, that are similar to liquid EO (Figure 7).

Spontaneous contraction is constant, contractility slightly modified without variability of spontaneous contractions, both for EO and for SEO. Excipients decrease contraction but not variability of contractions; therefore, the effect of essential oil on gallbladder are very small (Figure 8). In addition, the variability on FFT is observed only at higher concentrations (Appendix A).

#### Gastric Fundus

Liquid EO and excipients induce a modest decrease in the gastric fundus tone. SEO progressively decreases the gastric tone up to 70% at the highest studied dose (Figure 9).

Liquid EO, SEO, and excipients do not influence spontaneous contractility pattern in a concentration dependent manner (Figure 10). The pattern of the contractions is regularly maintained. EO tone remains constant and increases by 1 mg/ml concentration and then decreases. Compared to the control, the variability drops but shows a slight increase to 10 mg/mL concentration compared to the smaller ones (Figure 10). SEO tone gradually decreases; the variability increases for concentrations greater than 1 mg/mL (Figure 10). Excipients tone drops (Figure 10); the variability is constant up to 1 mg/ml, and then it increases to 10 mg/mL (Figure 10). In all case regarding the FFT, low frequency prevails (Appendix A).

#### 3.2.2. Induced Contractility

Liquid EO, SEO, and excipients have been studied on intestinal segments depolarized by high K^+^ (80 mM) to evaluate its spasmolytic effects by affecting calcium movements through L-type calcium channels.

##### Ileum and Colon

Liquid EO inhibition is independent on dose while on the contrary, SEO and excipients act in dose-dependent manner and present overlapping curves. EO inhibited the activity by 77.4 ± 0.2% already at the lowest concentration tested of 0.1 mg/mL. The same inhibition was attained by SEO and excipients at 50-fold and 100-fold higher concentration, respectively. (Figure 11A). The excipients, separately studied, have similar values of intrinsic activity: 72 ± 1.9 but at five times greater concentrations (Figure 11). The same trend is repeated in the colon (Figure 11C), with the difference that the excipients, separately taken, have a maximum intrinsic activity at 5 mg/mL with less potency (Table 2). OB taken as a positive control has higher spasmolytic potency on colon than ileum (IC_50_ ileum 8.3 µM, colon 4.8 µM). Although it is not possible to make quantitative comparisons, since otilonium is a single molecule and EO is a complex mixture of compounds, the activity profile of is very similar, suggesting that they probably act on the same targets.

Percent inhibition is independent on concentration for liquid EO and similar in the ileum and colon. Excipients present a similar concentration-dependent decrease, overlapping SEO in both organs, as the effect of excipients seems the same.

##### Gallbladder

The same study done on the gallbladder showed free EO and SEO action. The intrinsic activity is 84 ± 2.4 (0.1 mg/mL) and 77 ± 1.6 (10 mg/mL), respectively. As for potency, SEO is 30 times less potent than liquid EO. The excipients have no intrinsic activity worthy of note like the reference compound (Table 2). Interestingly, intrinsic spasmolytic activity of OB on gallbladder is negligible.

### 3.3. Antimicrobial Activity

In order to confirm the antibacterial activity of SEO, we have evaluated effects on some lines of bacteria and fungi. Table 3 shows minimal inhibitory concentration (MIC) values. As can be seen, the solid version maintains the bactericidal action against some pathogens taken as a reference while the excipients are devoid of effects as already documented in literature [20].

Unlike cyprofloaxacin, taken as positive control, SEO and free EO did not show any effect on *Bifido* and *Lactobacillus*. In addition, in negative control no growth inhibition was observed.

## 4. Discussion

The three pharmacologic agents currently indicated in the USA for treatment of irritable bowel syndrome with diarrhea (IBS-D) are non-systemic antibiotic rifaximin, the mixed µ- and κ-opioid receptor agonist/δ-opioid antagonist eluxadoline, and the selective serotonin 5-HT_3_ antagonist alosetron [21]. An acceptable initial therapy, especially for patients with mild disease, is lifestyle modification and education. In this context, the antispasmodic and spasmolytic action of a drug is used to treat excessive painful muscle contractility of the intestine [22]. However, loperamide, which inhibits peristalsis and increases colonic transit time, is not helpful with abdominal pain [23], and often more than a drug is necessary. The association of antispasmodic and antinociceptive activity should be important. Moreover, the available synthetic antispasmodic and/or spasmolytic molecules often present severe side effects, limiting treatment efficiency and patient compliance. Therefore, the pharmaceutical industry is now searching for developing new drug candidates from plants rich in essential oils [24]. EOs antioxidant, anti-inflammatory, and antitumoral effects are widely known, together with their antinociceptive activities; they act on the digestive system [25] and improve the digestion process by stimulating the olfactory nerve endings [26,27]. Their antispasmodic effect has been less investigated, mainly for their difficult oral administration and their local aggression, though they are used in worldwide medicine. Peppermint oil administered orally in an enteric coated form [28,29], was efficacious in reducing global symptoms and pain in IBS [30].

With this view, we have evaluated the effect of a solid formulation of *Thymus vulgaris* L. EO, that presents spasmolytic and nociceptive effects [31] on the modulation of guinea pig ileum and colon basal and induced contractility, in order to have experimental evidences of its antispasmodic and spasmolytic intestinal effect. The study of the contractility of gallbladder and gastric fundus has been done to rule out the possibility of side effects.

In Thymus EO, mainly phenolic compounds containing hydrogen, carbon, and oxygen are present. By applying the original CEC-DAD and LC-MS/MS methodologies developed ad hoc for this study to Thymus EO and to an EO-based formulation, it was observed how the main bioactive constituents of both considered samples are represented by thymol and carvacrol together with other minor constituents for a total of 12 selected analysed compounds, among monoterpenes, bicyclic monoterpenes, monoterpenols, bicyclic monoterpenols, and sesquiterpene lactones. Such results are consistent with the literature on the subject [32,33], also considering that the type of cultivar, the geographical area, and seasonality significantly influence the content of bioactive compounds in *Thymus* essential oil composition and thus its chemotype [34,35]. Based on these results, the most represented phytochemicals thymol and carvacrol are responsible for the modulation of contractility of EO and SEO. However, the interactions of different phytochemicals present in the phytocomplex could produce synergistic antispastic and spasmolytic effects observed in ex vivo experiments [36].

In the basal conditions, SEO, consisting in excipients associated to liquid EO, is less effective than liquid EO in reducing the muscular tone in both the ileum and colon; its effects are dose-dependent and comparable to OB. However, in the ileum, the association of the excipients to liquid EO permits the maintenance of a normal pattern of waves that are abolished by liquid EO, consistently with SCV results. Therefore, the solid based formulation can functionally be proposed as ileal antispastic. In the colon, EO and SEO decrease the basal tone, and their effect is twice that of OB. The contractions are almost abolished by EO and SEO, differently from OB, which maintains a normal contractility route. The fact that EO and SEO possess comparable effects is surprising as it should be considered that in SEO, which is given to humans, EO represents the 0.18 % w/w (i.e., 1 mg SEO contains 0.002 mg EO). This suggest that EO, when administered with excipients, exert consistent effects also in the micromolar range of concentration. Consistently, it has been reported [37] that a thyme extract possesses spasmolytic activity both on trachea and intestinal smooth muscle due to thymol and carvacrol, as shown by experiments with each molecule were tested separately. In that study, the activity was not directly proportional to concentration, since the lower doses were the most active. This observation is in agreement with the present data in which very low EO concentration are very active, especially when considering that the thymol and carvacrol concentration in the solid form are comparable to those of the above-described reported data [37]. Another possibility is that the phytocomplex and excipients cooperate. As excipients per se are poorly active, we can speculate that their components strongly potentiate the effects of EO; in the presence of excipients, in fact, a 500-fold lower EO amount elicits effects comparable to EO per se.

The effect of essential oil seems stronger in the colon than in the ileum: this fact is important because, in the ileum, the contractile activity is maintained, and the low frequency waves are present also if the intestinal tone is reduced. The presence of the basal contraction rate pattern allows the mixing of the internal luminal content that is specific of the small intestine and not typical of the colon. The areas of alternate contraction and stretching present segmentation that may be particularly important in securing mixing: a recent paper speculates that the timing of segmentation contractions is largely, if not entirely, the result of slow wave activity in the intestinal smooth muscle coat [38]. In the colon, the low frequency bands and the contractility pattern are really diminished leading to a low contractile activity that associated to the decreased tone, may be helpful in colonic diarrheal syndrome.

Regarding the L-type calcium channels effects, in the ileum, liquid EO presents high activity and high potency, SEO lower activity, and much lower potency than EO, probably due to excipients; in the colon, liquid EO shows the same activity as in the colon, but SEO shows less activity than in the ileum, and EO and SEO show half their respective potency than in the ileum. OB has the highest activity and potency in the colon with respect to the ileum. The ability of SEO to block calcium-mediated events in gastrointestinal smooth muscle would lead to a local reduction in ileal and intestinal muscle tone.

A possible direct modulation of the formulation on the L-type calcium channels on the self-excitable cells of the ileum and colon opens an interesting set of potential targets for its activity. We can speculate, in fact, that thyme oil affects indirectly the mechanisms which drive motility trough membrane receptors, the activation of which is linked to the entry of calcium into the cell. Moreover, the possibility that the phytocomplex could directly bind other receptors involved in gut motility, as already demonstrated for the cholinergic receptor [36], cannot be ruled out. To the SEO, spasmolytic action also contributes to monoterpenes, for which antispasmodic activity has already been shown [39]. In particular, in SEO, there is an interesting amount of *p*-cymene (Table 1), to which the literature attributes antispasmodic action through interaction with receptors directly involved in the control of motility such as cholinergic ones [40].

In addition, we have studied gallbladder and gastric fundus contractility as off target districts. EO and SEO do not modify the spontaneous basal contraction of gallbladder; on calcium induced contractility, EO exerts a relaxant activity on gallbladder, but the solid formulation, although maintaining a relaxant activity on ileum and colon, reduces the potency by four and two times on gallbladder with respect to the ileum and colon (Table 2).

The antibacterial activity of essential oils has long been known [41] and seems to be linked to the prevalent chemical chemotype, with phenols being the most active compounds. The scientific community is in agreement that the actions of these natural phytocomplexes depend not only on the compounds present in greater quantities but on the chemotype. Indeed, it is possible to find very powerful actions [42,43] but also phytocomplexes with much lower antimicrobial action [44,45]. The Thymus oil used contains predominantly phenolic monoterpenes (Table 1) and maintains its antibacterial action even in the solid form by selectively acting on pathogenic bacteria (Table 3). It is interesting to underline how the formulation maintains action in line with liquid oil even if with lower power. Both are without effects on commensal bacteria. This data is particularly interesting for the importance of the microbiota in intestinal homeostasis [46] and particularly for the strong action on *Streptococcus pyogenes* (gram^+^) and *Pseudomonas aeruginosa* (Gram^−^). The same can be said for *Candida albicans* (Table 3). Carvacrol and thymol, being hydrophobic, can interfere with the lipid bilayer of cytoplasmic membranes of bacteria, bringing loss of integrity and increasing its fluidity and permeability and leakage of cellular material such as ions [47]. Biologically active molecules probably maintain the ability to pass the bacterial wall, enter the cytoplasm and perform their bactericidal action compromising the vital functions of the bacterium itself. This action is also described in the literature for thymol which represents the prevalent compound [48] and for *p*-cymene, which has been proven to possess interesting in vitro antimicrobial activity [40].

In conclusion, our work focuses mainly on spontaneous contractility by highlighting an interesting activity profile of SEO. This is probably due to the direct action of the formulation on the L-type calcium channels on the self-excitable cells of the ileum and colon. The possible modulation of L-type calcium channels opens up an interesting set of potential targets for its activity. In conclusion, this formulation probably modulates various nodes of the target network connected to diarrhoea owing to spasmolytic and antispasmodic action on ileum and colon. The solid form allows systemic applications and makes it possible for use in systemic diseases. In addition, the anti-tumour action of essential oil demonstrated for some cell lines [49] can be an interesting added value.

## Figures and Tables

**Figure 1 biomolecules-10-00860-f001:**
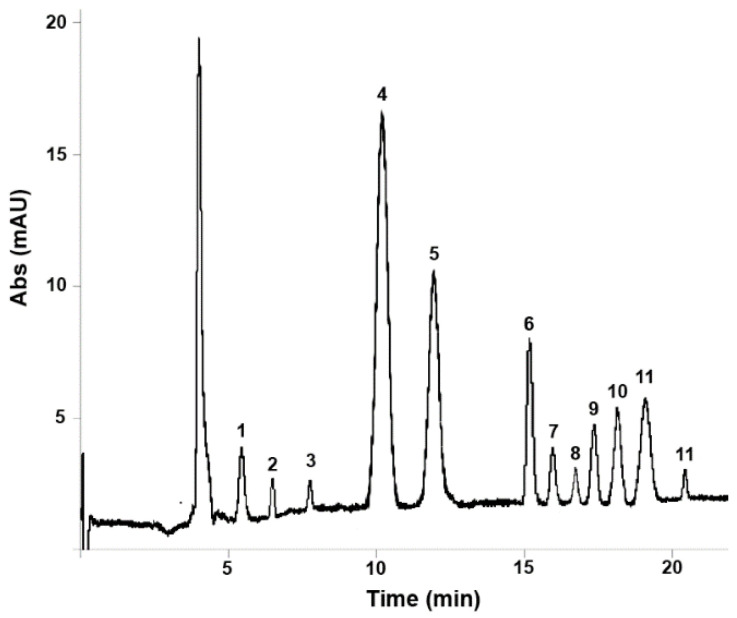
Capillary electrochromatography coupled to diode array detection (CEC-DAD) electrochromatogram obtained from the analysis of a *Thymus vulgaris* L. essential oil sample under the optimised conditions: 1, borneol; 2, linalool; 3, α-terpineol; 4, thymol; 5, carvacrol; 6, p-cymene; 7, β-pinene; 8, α-terpinene; 9, β-myrcene; 10, β-Cariophyllene; 11, γ-terpinene; 12, limonene.

**Figure 2 biomolecules-10-00860-f002:**
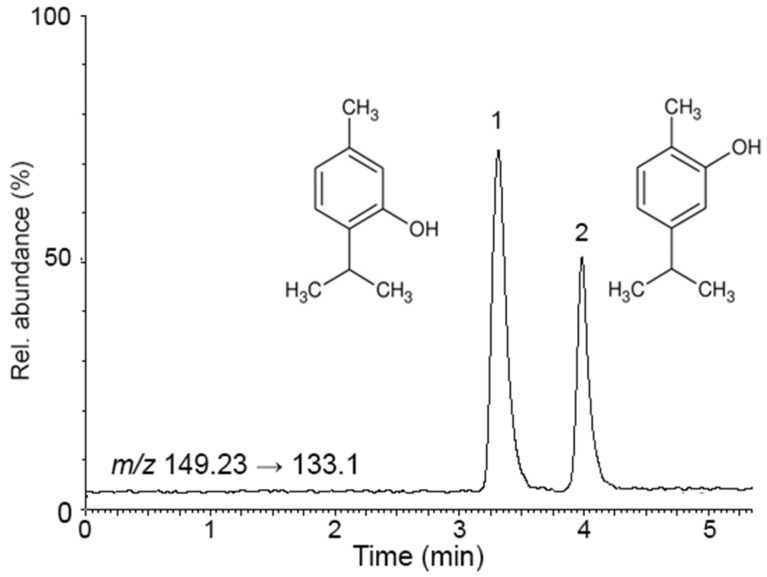
Liquid chromatography-tandem mass spectrometry (LC-MS/MS) chromatogram obtained from the analysis of a *Thymus vulgaris* L. essential oil–based formulation sample under the optimised conditions: 1, thymol; 2, carvacrol.

**Figure 3 biomolecules-10-00860-f003:**
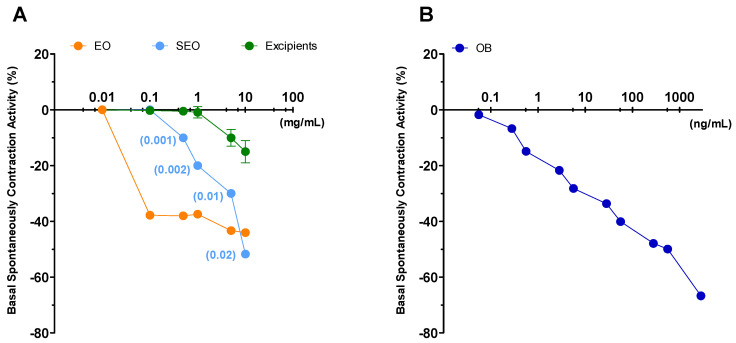
Ileum: basal spontaneous contraction activity. Zero represents the basal tone and each point is the percent variation from the baseline after cumulative addition of each dose. (**A**) Essential oil (EO), solid essential oil (SEO), and excipients (mg/mL); numbers in brackets represent the effective EO concentration (mg/mL) in SEO; (**B**) otilonium bromide (OB) (ng/mL), an antispasmodic drug, used as a positive control. Each value (expressed as percent variation) is the mean ± SEM; when the error bar is not shown, it is covered by the point.

**Figure 4 biomolecules-10-00860-f004:**
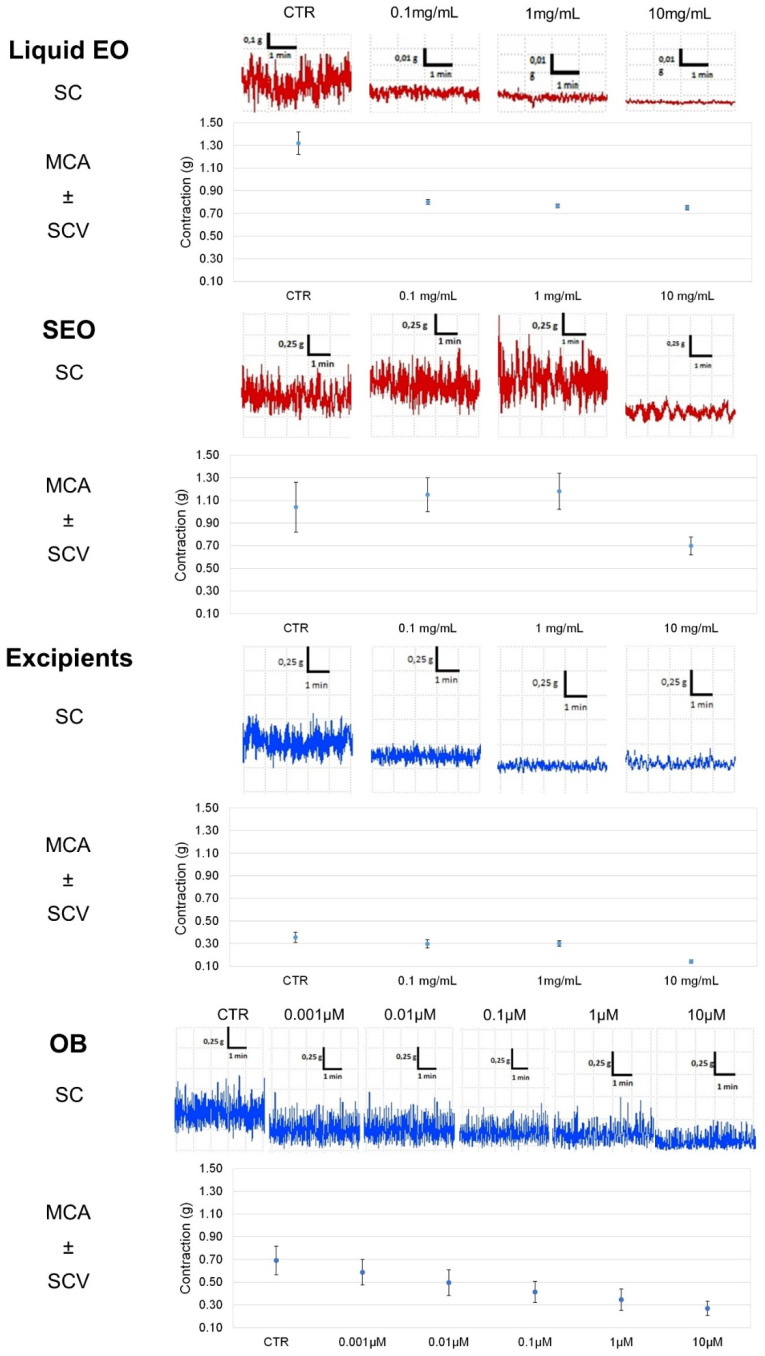
Focus on experimental original recording of the concentration-response curve of EO and SEO, excipients, and otilonium bromide (OB) on spontaneous ileum basal contractility. Spontaneous contraction (SC); spontaneous contraction variability (SCV); mean contraction amplitude (MCA).

**Figure 5 biomolecules-10-00860-f005:**
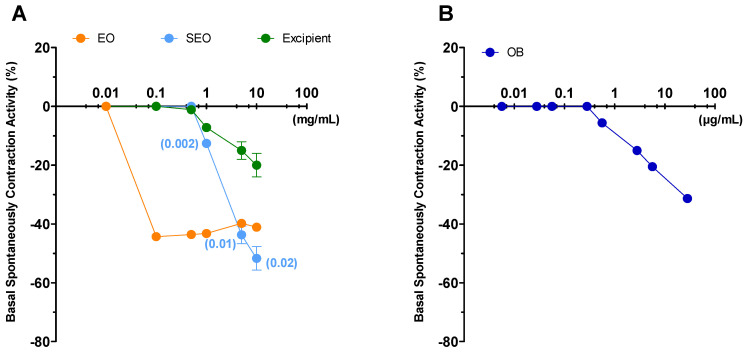
Basal spontaneous contraction activity (BSCA) in the colon. Zero represents the basal tone and each point is the percent variation from the baseline after cumulative addition of each dose. (**A**) EO, SEO and excipients (mg/mL); numbers in brackets represent the effective EO concentration (mg/mL) in SEO. (**B**) Otilonium bromide (OB) (µg/mL), an antispasmodic drug, used as a positive control. Each value (expressed as percent variation) is the mean ± SEM; when the error bar is not shown, it is covered by the point.

**Figure 6 biomolecules-10-00860-f006:**
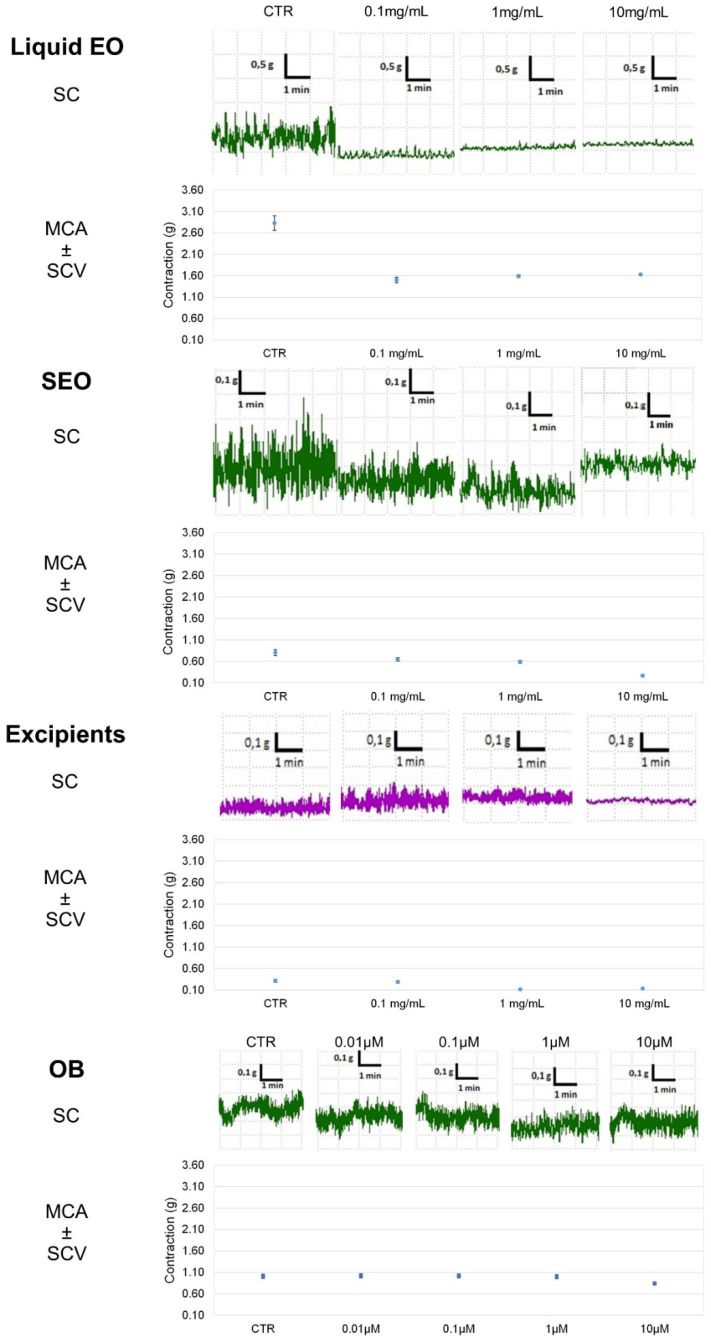
Focus on experimental original recording of the concentration-response curve of liquid EO and SEO, excipients, and OB on spontaneous colon basal contractility. Spontaneous Contraction (SC); spontaneous contraction variability (SCV); mean contraction amplitude (MCA). Absolute band powers of control and after addition of each concentration observed in the same experiment.

**Figure 7 biomolecules-10-00860-f007:**
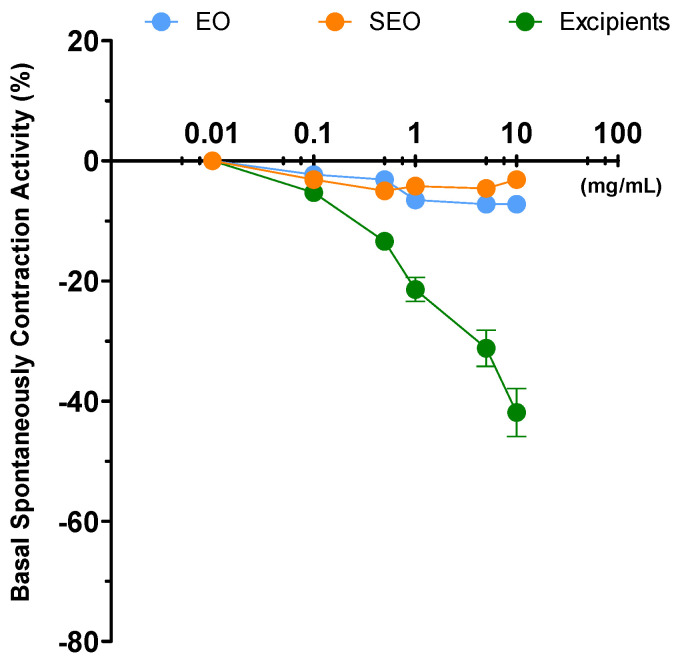
Basal spontaneous contraction activity elicited by free EO, SEO, and excipients in the gallbladder. Zero represents the basal tone and each point is the percent variation from the baseline after cumulative addition of each dose. Each value (expressed as percent variation) is the mean ± SEM; when the error bar is not shown, it is covered by the point.

**Figure 8 biomolecules-10-00860-f008:**
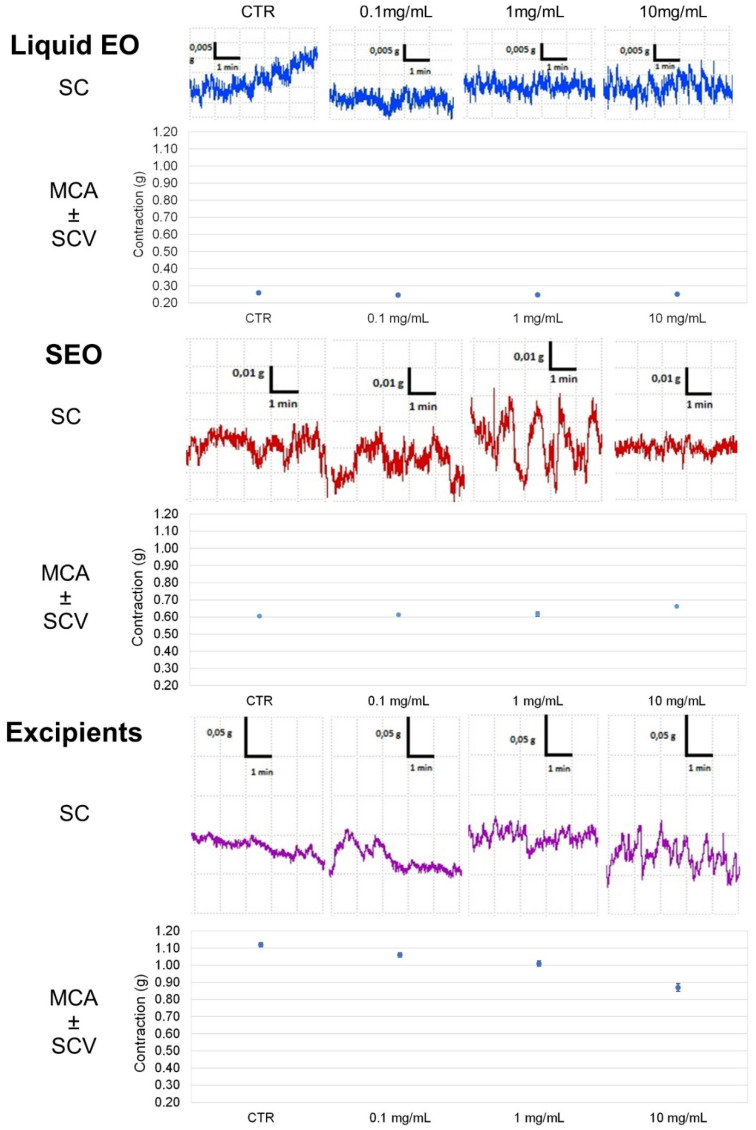
Focus on experimental original recording of the concentration-response curve of liquid EO, SEO, and excipients on spontaneous gallbladder basal contractility. Spontaneous contraction (SC); spontaneous contraction variability (SCV); mean contraction amplitude (MCA).

**Figure 9 biomolecules-10-00860-f009:**
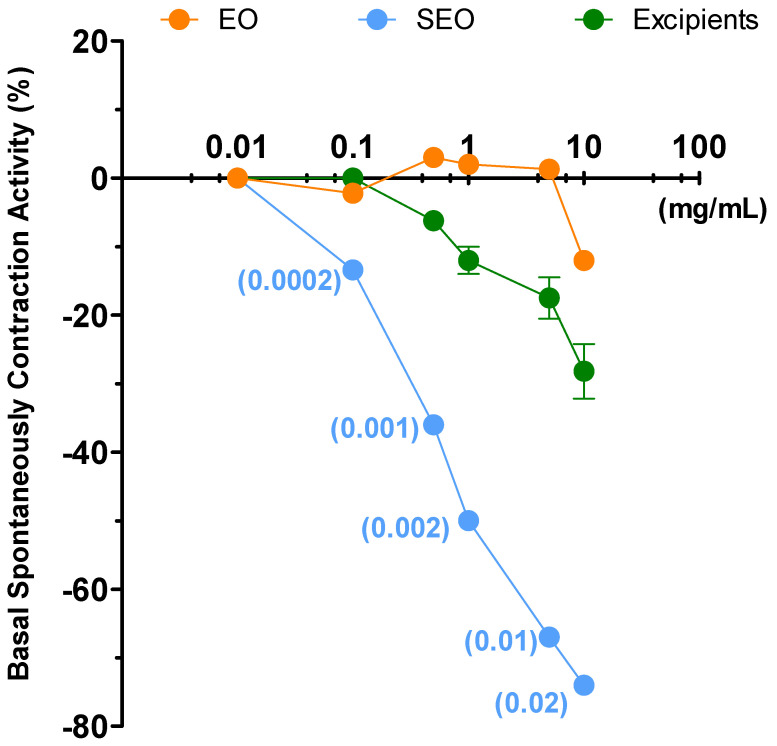
Basal spontaneous contraction activity elicited by free EO, SEO, and excipients in the gastric fundus. Zero represents the basal tone and each point is the percent variation from the baseline after cumulative addition of each dose. Numbers in brackets represent the effective EO concentration (mg/mL) in SEO. Each value (expressed as percent variation) is the mean ± SEM; when the error bar is not shown, it is covered by the point.

**Figure 10 biomolecules-10-00860-f010:**
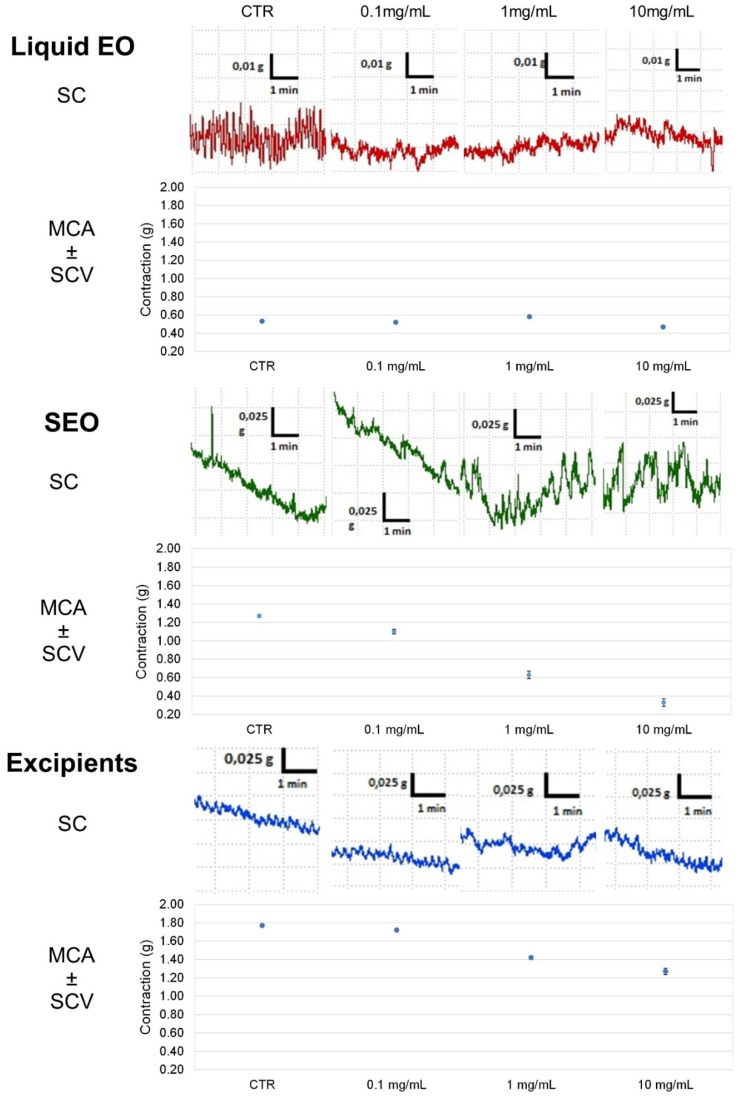
Focus on experimental original recording of the concentration-response curve of liquid EO, SEO and excipients on spontaneous gastric fundus basal contractility. Spontaneous contraction (SC); spontaneous contraction variability (SCV); mean contraction amplitude (MCA).

**Figure 11 biomolecules-10-00860-f011:**
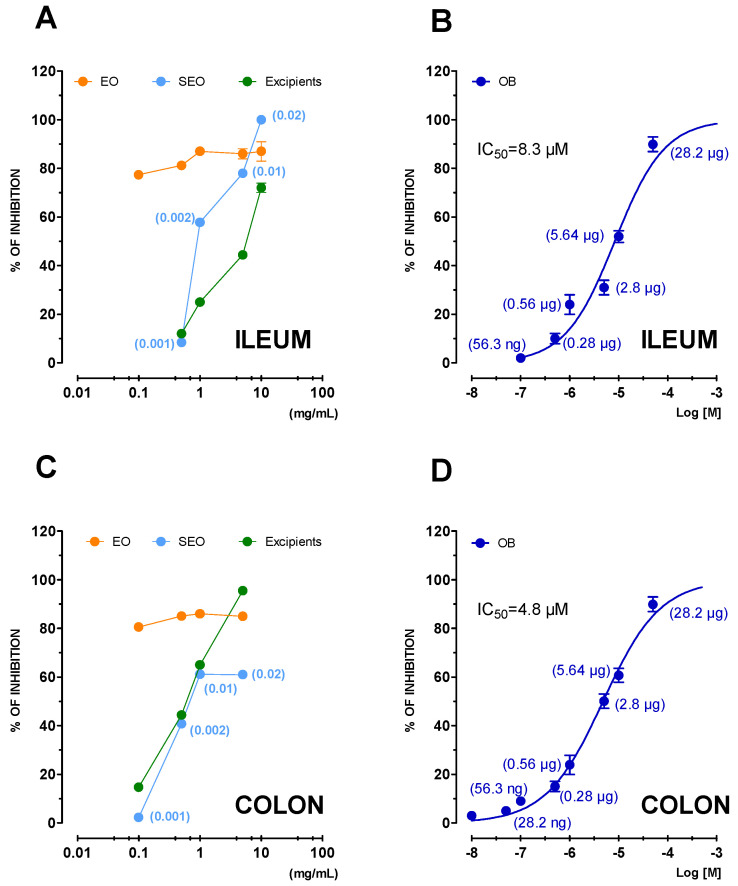
Cumulative concentration-response curves of spasmolytic activity of *Thymus vulgaris* L. free EO, SEO, and excipients and otilonium bromide (OB) against potassium chloride- (80 mM) induced contraction on guinea pig ileum (panels **A** and **B**) and colon (panels **C** and **D**). Each point is the mean ± SEM of four-six experiments. Where error bars are not shown these are covered by the point itself. For a better comparison of the effects, numbers in brackets represent the effective EO concentration (mg/mL) in SEO (panels **A** and **C**) or the amount of OB present in 1 mL (panels **B** and **D**).

**Table 1 biomolecules-10-00860-t001:** Chemical composition of *Thymus vulgaris* L. essential oil and derived formulation.

Class	Compound	*Thymus vulgaris* L. EO (µg/100 µL)	EO-Based Formulation (µg/cps)
Monoterpenes	*p*-Cymene	9.4 ± 0.6	41.8 ± 6.5
	α-Terpinene	0.7 ± 0.2	4.4 ± 0.5
	γ-Terpinene	4.0 ± 0.7	19.3 ± 1.0
	β-Myrcene	2.1 ± 0.4	12.1 ± 0.8
	Limonene	0.4 ± 0.1	1.8 ± 0.3
Bicyclic monoterpenes	β-Pinene	1.1 ± 0.2	4.3 ± 0.6
Monoterpenols	Thymol	43.3 ± 1.4	210.2 ± 5.6
	Carvacrol	20.7 ± 2.3	99.7 ± 7.4
	Linalool	0.7 ± 0.2	3.4 ± 0.6
	α-Terpineol	0.2 ± 0.1	0.9 ± 0.2
Bicyclic monoterpenols	Borneol	1.3 ± 0.4	6.4 ± 0.7
Sesquiterpene lactones	β-Cariophyllene	3.1 ± 0.5	13.9 ± 1.0

**Table 2 biomolecules-10-00860-t002:** Relaxant activity of tested samples on K^+^-depolarized guinea pig intestinal smooth muscle.

		Activity *^a^*	Potency *^b^*	
Tissue	Comp.	M ± SEM	IC_50_	95% conf lim
**Ileum**	**Liquid EO**	87 ± 1.6 (*1)*		
	**SEO**	100 ± 1.3 (*5)*	1.12	1.02–1.47
	**Excip**	72 ± 1.9 (*5)*	1.64	1.36–1.98
	**OB**	90 ± 3 *(0.005)*	0.0048	0.0040–0.0057
**Colon**	**Liquid EO**	85 ± 1.9 (*0.5)*	0.031	0.009–0.043
	**SEO**	61 ± 2.4 (*1)*	0.70	0.57–0.85
	**Excip**	95 ± 1.7 (*5)*	1.33	0.99–1.48
	**OB**	90 ± 2.3 *(0.02)*	0.0019	0.0015–0.0025
**Gallbladder**	**Liquid EO**	84 ± 2.4 (*0.1)*	0.048	0.035–0.055
	**SEO**	77 ± 1.6 (*10)*	1.44	1.06–2.03
	**Excip**	38 ± 2.6 (*1)*		
	**OB**	10 ± 0.7 *(0.02)*		

*^a^* Percent inhibition of calcium-induced contraction on K^+^-depolarized (80 mM) guinea pig ileum, colon, and gallbladder. In parenthesis the indicated concentrations that give the maximum effect expressed as mg/mL. *^b^* IC_50_, expressed as mg/mL, represent the concentration that inhibited 50% of the maximum contraction induced by K^+^ 80 mM and was calculated from concentration-response curves (probit analysis by Litchfield and Wilcoxon [17] with *n* = 6–7).

**Table 3 biomolecules-10-00860-t003:** Antimicrobial activity of tested samples.

Microorganism Strain	MIC *^a^* (mg/mL)			
	Liquid EO	SEO	Excipient	Cyprofloaxacin
**Gram^+^ bacteria**				
*Staphylococcus aureus*	0.28	>50	Inactive	0.005
*Streptococcus pyogenes*	0.004	2	Inactive	0.002
*Bifidobacterium Breve*	Inactive	Inactive	Inactive	0.005
*Lactobacillus Fermentum*	Inactive	Inactive	Inactive	0.005
**Gram^−^ bacteria**				
*Pseudomonas aeruginosa*	0.0002	0.1	Inactive	0.0004
*Escherichia coli*	0.4	>50	Inactive	0.005
*Salmonella Thyphimurium*	0.33	>50	Inactive	0.005
**Fungus**				
*Candida albicans*	0.0018	9	Inactive	0.005
10% DMSO	Inactive	Inactive	Inactive	Inactive

*^a^* Minimal inhibition concentration (MIC) values.

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
