# Peer review of "Thymus vulgaris L. Essential Oil Solid Formulation: Chemical Profile and Spasmolytic and Antimicrobial Effects"

_biomolecules, 2020, doi:10.3390/biom10060860_

Round 1
Reviewer 1 Report
Comments to authors
In this work, the authors report the phytochemical characterization (CEC-DA, LC-MS/MS) of thyme essential oil in its raw liquid (EO) and encapsulated-diluted (SEO; 0.18% w/w) and their antispasmodic activity in certain GI organs (jejune, ileum, gallbladder, and gastric fundus) in an ex vivo bioanalytical system (organ bath). The chemical composition of EO was different than that reported by other authors, particularly in its carvacrol (PubChem CID: 10364) content. As expected, the physiological action was dose-dependent (EO>SEO) without significant interference from the components of the vehicle in which the oil was dissolved. EO and SEO had a lesser spasmolytic/antispastic activity than otilonium bromide (OB) but a lower antibacterial activity than previously reported by other authors for almost the same bacterial strains (PMID: 25870697; DOI: 10.1002/fsn3.1007).
The manuscript represents an advance on the field but major changes must be made before the manuscript is accepted for publication; the authors are asked to modify the following to improve its quality and scientific soundness
General
What is the rationale behind comparing thyme essential oil in its raw liquid (EO) and encapsulated-diluted (SEO; 0.18% w/w) forms? Particularly, if the excipient is inert in the assayed bioactivity?. Authors should reconfigure their results and discussion sections, reflecting such a “dilution effect”. After explaining in depth this issue, authors must reconsider SEO abbreviation since the pharmaceutical formulation seems to be a solid tablet.
Also, the English syntax is poor; it is requested that the manuscript be reviewed once again by a native of this language. Here’s an example: In traditional medicine, humans have [been] greatly benefited from plants’ secondary metabolites
Title. Should be much broader. Suggestion: Thymus vulgaris' essential oil-based caps: Chemical profile and spasmolytic & antibiotic effect.
Abstract. Should be described in a more quantitative (include p-values) and comparative manner. For example, Also, include information on the antibiotic activity.
Introduction. Line 57-58, “The oil was adhered to a solid structure and inserted into an operculum” – what do you mean?, Wasn’t an ex vivo study?. You also stated that “we have evaluated a formulation of SEO obtained through the adsorption onto a solid matrix” and such evaluation of pharmaceutical properties was not included in this study. Authors should put more emphasis on the scientific contribution (what's new) of their work to the current knowledge.
Materials and methods
Section 2.1- This section must be renamed as follows: “Thyme essential oil (EO) and pharmaceutical preparation”; authors must provide more details as to how they prepared the capsules which, by the way, weren’t they tablets?. The paragraph from lines 85 to 99 should be a separate section and placed before the current Section 2.1. See sections 2.1 and 2.2 in the following manuscript (DOI: 10.1002/fsn3.1007)
Section 2.2 - Because methodological differences between the two chromatographic methods are not fully discussed within the text, the use of LC-MS / MS should be considered an external validation method for CEC-DAD. Taking this into account, the methodological description of CEC-DAD (lines 101-108) should only be included in this section (as section 2.2.1.) and that of LC-MS / MS (lines 109-124) should be included as supplementary material. The same goes for section 2.2.2 (method validation). After having done this, section 2.2.3 (sample analysis) should be merged with this same section, including additional information such as analytical units (ug/cap?), the number of assays (# of batches, and replicates) from which average values and standard deviation (Table 3) were calculated from.
Section 2.3 - This seems to be the cornerstone of your research and therefore the detailed description and illustration should be improved. This section must be termed “ex vivo muscle contractibility evaluations”. A detailed description of the ex vivo biosystem (“organ bath”) including a picture or drawing revealing specific details of it (e.g. https://en.wikipedia.org/wiki/Organ_bath) is strongly suggested.
Section 2.4 – A more comprehensive description of antibiotic assays is needed. The authors can look up to the following (examples) to reconstruct this section: https://www.ncbi.nlm.nih.gov/pmc/articles/PMC4391421/ and https://onlinelibrary.wiley.com/doi/full/10.1002/fsn3.1007 (Section 2.8). Please include data from Supplementary Material (S2).
- Results. Eliminate the paragraph located between lines 196-198. It is the opinion of this reviewer that sections 3.1.1 and 3.1.2 should eliminate as they correspond to methods’ standardization, and not results per see; instead, please include method reliability data in only one or two lines before discussing section 3.1.3. Section 3.2 should be reconstructed in such a way that all differences between SEO versus EO and OB become more evident (clear).
- Figures. Too many figures. Figures 1 and 2 can be either eliminated or included as supplementary material since they correspond to method standardization. Graphs A (EO and SEO) and B (OB) from Figures 3 and 5 and graphs A-D in Figure 11 should have the same units, regardless of magnitude differences. All other figures should be eliminated if not discusses properly, and all should be provided with a higher resolution (>300 dpi).
- Tables. Table 1 can be deleted and just described within the text; by the way, it seems more a tablet rather than a capsule formulation. Table 2 can be either eliminated or included as supplementary material since it corresponds to method standardization; is CEC-DAD in fact p-Cymene in the analyte column?. Table 3, LC-MS/MS data can be eliminated and just discussed as a footnote. EO (% m/V) and SEO (ug/cps) should have the same units. Tables 4 and 5 please review common format (instruction for authors).
- Discussion. Too short, It should be ore comparative with preceding studies on chemical characterization and antibiotic activity (DOI: 10.7324/JABB.2017.50203, 10.1016/j.foodres.2017.03.006, 10.1002/fsn3.1007; PMID: 25870697) and antispasmodic natural extracts, including thyme (http://journal.skums.ac.ir/article-1-610-en.html; http://eprints.skums.ac.ir/934/)
- References. Certain references are not formatted according to biomolecules. Improve the number of references with less of 10y old up to 80% (currently 32% are ≥10 y old)
Author Response
Bologna, May 25, 2020
Revision of Manuscript ID: biomolecules-798449
“Thymus vulgaris L. essential oil-based formulation: chemical analysis and in vitro intestinal contractility study”
By Matteo Micucci, Michele Protti, Rita Aldini, Maria Frosini, Ivan Corazza, Carla Marzetti, Laura Beatrice Mattioli, Gabriella Tocci, Alberto Chiarini, Laura Mercolini and Roberta Budriesi
Dear Editor,
We have carefully read the comments of the Reviewers and we are grateful for the useful criticisms that allowed us to improve the quality of this study. The manuscript has been revised, and we hope that in the present form it is suitable for publication.
Below we report our answers to specific point raised by the Reviewers.
Thank you again for your attention and we look forward to hearing from you soon.
Yours faithfully,
Roberta Budriesi, Prof.
|
Reviewer 1: |
|
• Comments to authors In this work, the authors report the phytochemical characterization (CEC-DA, LC-MS/MS) of thyme essential oil in its raw liquid (EO) and encapsulated-diluted (SEO; 0.18% w/w) and their antispasmodic activity in certain GI organs (jejune, ileum, gallbladder, and gastric fundus) in an ex vivo bioanalytical system (organ bath). The chemical composition of EO was different than that reported by other authors, particularly in its carvacrol (PubChem CID: 10364) content. As expected, the physiological action was dose-dependent (EO>SEO) without significant interference from the components of the vehicle in which the oil was dissolved. EO and SEO had a lesser spasmolytic/antispastic activity than otilonium bromide (OB) but a lower antibacterial activity than previously reported by other authors for almost the same bacterial strains (PMID: 25870697; DOI: 10.1002/fsn3.1007). We agree with what the Referee says about the different powers on the studied actions of our essential oil. As for otilonium bromide, which we have taken as a positive control, we are well aware of the different power: this is a drug while liquid or solid essential oil are a mixture of compounds. In our opinion, the important thing is the ability of the oil in both versions to bind targets linked to the intestinal motor pathology and therefore be useful as an adjuvant (and not as a drug) even if with a quantitatively lower action. Another goal of this work has been to verify a possible use of this solid version as a supplement in systemic diseases. This version bypasses the disadvantages of the liquid version oil. We also agree with the referee that the antimicrobial activities that we find are lower than those reported in the papers suggested [Thymus vulgaris essential oil: chemical composition and antimicrobial activity O Borugă, C Jianu, C Mişcă, I Goleţ, AT Gruia, FG Horhat J Med Life. 2014; 7(Spec Iss 3): 56–60. (PMID: 25870697)] [Evaluation of Thymus vulgaris and Thymbra spicata essential oils and plant extracts for chemical composition, antioxidant, and antimicrobial properties Ayça Gedikoğlu, Münevver Sökmen, Ayşe Çivit Food Sci Nutr. 2019 May; 7(5): 1704–1714. (DOI: 10.1002/fsn3.1007)]. The scientific community is quite in agreement that the actions of these mixtures of compounds do not depend only on the concentration of the prevailing compounds (in this case thymol and carvacrol), but also on all the other compounds present in the mixture and by their concentration: therefore the chemotype influences the various activities. Indeed, much lower actions can also be found [Cáceres M, Hidalgo W, Stashenko E, Torres R, Ortiz C. Essential Oils of Aromatic Plants with Antibacterial, Anti-Biofilm and Anti-Quorum Sensing Activities against Pathogenic Bacteria. Antibiotics (Basel). 2020 Mar 30;9(4). pii: E147.] [Kerekes EB, Vidács A, Takó M, Petkovits T, Vágvölgyi C, Horváth G, Balázs VL, Krisch J. Anti-Biofilm Effect of Selected Essential Oils and Main Components on Mono- and Polymicrobic Bacterial Cultures. Microorganisms. 2019 Sep 12;7(9). pii: E345]. Therefore, the differences obtained in our study can be attributed to the different chemical composition in quantitative terms of the phytocomplex. However, in accordance with what the Referee suggested, the discussion about antimicrobial action has been implemented. The manuscript represents an advance on the field but major changes must be made before the manuscript is accepted for publication; the authors are asked to modify the following to improve its quality and scientific soundness. We thank the referee for the appreciation about the paper and we have made the suggested changes. General What is the rationale behind comparing thyme essential oil in its raw liquid (EO) and encapsulated-diluted (SEO; 0.18% w/w) forms? The rationale is obtain info about the contractility activity of the form in this formulation, because encapsulation of solid essential oils is an efficient approach to overcome the distaste, modulate drug release, increase the stability of the active compounds, inhibit volatility, toxicity and interactions with the intestinal substances and improve patient compliance and convenience. Particularly, if the excipient is inert in the assayed bioactivity? Authors should reconfigure their results and discussion sections, reflecting such a “dilution effect”. After explaining in depth this issue, authors must reconsider SEO abbreviation since the pharmaceutical formulation seems to be a solid tablet. The pharmaceutical formulation is contained in a solid capsule, but we have studied the formulation inside the solid capsule, which protects the formulation from the gastric juice. The capsule has been discharged. The effect of the excipients used in the formulation of the solid oil has been also studied in the ex vivo studies: their activity is consistent with the active essential oil, but at much higher concentration. We cannot say that they are inert, but they do not impair the activity. In the studied formulation, Thymus vulgaris L. is present at the value of 0.18 % w/w of the essential liquid oil as under patent. The dilution effect is reconfigured and discussed. The formulation is not a solid tablet. We think we can maintain the abbreviation SEO. Also, the English syntax is poor; it is requested that the manuscript be reviewed once again by a native of this language. Here’s an example: In traditional medicine, humans have [been] greatly benefited from plants’ secondary metabolites The English language has now been carefully revised throughout the manuscript in terms of syntax. Title. Should be much broader. Suggestion: Thymus vulgaris' essential oil-based caps: Chemical profile and spasmolytic & antibiotic effect. We have accepted in part the Referee's suggestion: the title has been changed as follows: Thymus vulgaris L. essential oil solid formulation: chemical profile, spasmolytic and antimicrobial effects. In fact, we have used the powder inside the caps, that is the solid based formulation, not the caps. Abstract. Should be described in a more quantitative (include p-values) and comparative manner. For example, Also, include information on the antibiotic activity. Only a qualitative comparison was possible, most important for a possible therapeutic use. P-values are not reported. Introduction. Line 57-58, “The oil was adhered to a solid structure and inserted into an operculum” – what do you mean? Wasn’t an ex vivo study? You also stated that “we have evaluated a formulation of SEO obtained through the adsorption onto a solid matrix” and such evaluation of pharmaceutical properties was not included in this study. Authors should put more emphasis on the scientific contribution (what's new) of their work to the current knowledge. “The oil was adhered to a solid structure and inserted into an operculum”: the essential oil was adhered to a solid structure: this form was studied for the contractility studies, before being placed in the operculum. In the paper we describe the procedure by which we have obtained the adsorption of the essential oil onto a solid matrix”. The operculum was discharged. The evaluation of pharmaceutical properties of the formulation are described (Introduction) and the scientific contribution of the work is reported. Lines 58-65: In this paper we have studied a peculiar formulation of Thymus vulgaris L. EO absorbed to a solid matrix inserted into a capsule for a potential use in intestinal pathologies. The operculum was discharged and the solid liquid oil was obtained. The rationale of this study was to obtain information about the effect on intestinal contractility of this solid form because this approach can overcome the distaste, modulate drug release, increase the stability of the active essential oil, can inhibit volatility, toxicity and interactions with the intestinal substances and improve patient compliance and convenience. The effect of the solid form has been compared with the liquid essential oil and the excipients. Materials and methods. Section 2.1- This section must be renamed as follows: “Thyme essential oil (EO) and pharmaceutical preparation”; authors must provide more details as to how they prepared the capsules which, by the way, weren’t they tablets? The paragraph from lines 85 to 99 should be a separate section and placed before the current Section 2.1. See sections 2.1 and 2.2 in the following manuscript (DOI: 10.1002/fsn3.1007). The Thymus vulgaris L. solid essential oil formulation named Aromatoil® [manifactured by Coima, Bastia (RA), Italy] used in this study was supplied by BIO-LOGICA, Bologna. Solid based Thymus vulgaris pharmacological composition was: 0.6 mg essential oil and 340.4 mg Excipients (pregelatinized corn starch, soy lecithin, ascorbic acid, calcium carbonate, levilite, vegetable magnesium stearate) The used essential oil has been obtained by steam distillation of the summit flowers. Section 2.2 - Because methodological differences between the two chromatographic methods are not fully discussed within the text, the use of LC-MS / MS should be considered an external validation method for CEC-DAD. Taking this into account, the methodological description of CEC-DAD (lines 101-108) should only be included in this section (as section 2.2.1.) and that of LC-MS / MS (lines 109-124) should be included as supplementary material. The same goes for section 2.2.2 (method validation). After having done this, section 2.2.3 (sample analysis) should be merged with this same section, including additional information such as analytical units (ug/cap?), the number of assays (# of batches, and replicates) from which average values and standard deviation (Table 3) were calculated from. Following the Reviewer’s suggestion, LC-MS/MS system description has now been included as Supplementary Material (S1), while sections 2.2.2 and 2.2.3 have been merged as section 2.2.2 Sample analysis, now including the requested additional information. Section 2.3 - This seems to be the cornerstone of your research and therefore the detailed description and illustration should be improved. This section must be termed “ex vivo muscle contractibility evaluations”. A detailed description of the ex vivo biosystem (“organ bath”) including a picture or drawing revealing specific details of it (e.g. https://en.wikipedia.org/wiki/Organ_bath) is strongly suggested. If the Reviewer asks for a more complete description of the method, we can add in the supplementary a detailed description or refer to a previous paper. We can add in the supplementary also the following paragraph: In a typical organ bath preparation, we put an excised piece of smooth muscle tissue (or cardiac tissue) in a chamber oxygenated with carbogen and kept in a solution such as Tyrode's solution. The tissue is attached to a lever, which transmits its contraction to a myograph and the physiological response is registered. Substances that are investigated are put directly to the chamber. We are studying the contraction of smooth muscle in tissues such as ileum, colon, trachea, bladder and blood vessels such as aortic rings and of the heart atria and ventricula. The main advantage of this technique is that the tissue lives and functions as a whole, with a physiological outcome (contraction or relaxation). It is a synthesis of steps (drug-receptor interaction, signal transduction, second generation messenger, change in excitability of smooth muscle). While other techniques allow the study of each of these steps, isolated tissue integrates all these steps. Moreover, Another important pharmacological show how the drugs tested would work in the body as a whole. A segment of the small intestine freshly dissected. A tension transducer (range of 0–50 g) to connect to a computer with basic data-acquisition software. This should be capable of displaying a simple graph of tension against time and marking drugs. There are commercially available computer-based converter and display systems. Currently, we use equipment and software that have been manufactured on site and adapted for display on personal computers. An organ bath (with a 50-ml level indicator) to support the tension transducer and keep the tissue in a warm oxygenated environment. The temperature of the organ bath is set at 37°C. A combination of thread and spring clips to connect the segment of the gut to the tension transducer. A picture of our recording data apparatus has been added to the Supplementary Material file. Section 2.4 – A more comprehensive description of antibiotic assays is needed. The authors can look up to the following (examples) to reconstruct this section: https://www.ncbi.nlm.nih.gov/pmc/articles/PMC4391421/ and https://onlinelibrary.wiley.com/doi/full/10.1002/fsn3.1007 (Section 2.8). Please include data from Supplementary Material (S2). We apologize to the referee: we failed to mention one of our bibliography on the method used to evaluate the antimicrobial effect Supplementary Material (S2). We have added an appropriate reference. If the referee considers it appropriate we can put all the information concerning the method in the paper even if we think it should be inserted in Supplementary material. Results. Eliminate the paragraph located between lines 196-198. It is the opinion of this reviewer that sections 3.1.1 and 3.1.2 should eliminate as they correspond to methods’ standardization, and not results per see; instead, please include method reliability data in only one or two lines before discussing section 3.1.3. As suggested by the Reviewer, sections 3.1.1 and 3.1.2 have been eliminated from the manuscript and included in the Supplementary Material (S4), while a brief summary of method standardisation (method validation) results has been included before method application to samples. Section 3.2 should be reconstructed in such a way that all differences between SEO versus EO and OB become more evident (clear). Section 3.2 has been completely revised as suggested by the referee. Figures. Too many figures. Figures 1 and 2 can be either eliminated or included as supplementary material since they correspond to method standardization. The Authors would like to better clarify that Figures 1 and 2 refer respectively to the CEC-DAD electrochromatogram and the LC-MS/MS chromatogram deriving from method application to real samples, which therefore correlate with quantitative results shown in Table 2. For this reason, in the Author’s opinion, these figures are not referred to method standardisation (method validation) but represent the core of analytical characterization results, and for this reason the Authors believe it is important to keep them in the manuscript. Graphs A (EO and SEO) and B (OB) from Figures 3 and 5 and graphs A-D in Figure 11 should have the same units, regardless of magnitude differences. All other figures should be eliminated if not discusses properly, and all should be provided with a higher resolution (>300 dpi). Tables.
rather than a capsule formulation. In agreement with referee, Table 1 has been deleted: it is the content of the capsule, not the capsule. It is not a tablet: inside the operculum it was comtained: Solid based Thymus vulgaris pharmacological composition was: 0.6 mg essential oil and 340.4 mg Excipients (pregelatinized corn starch, soy lecithin, ascorbic acid, calcium carbonate, levilite, vegetable magnesium stearate.
to method standardization; is CEC-DAD in fact p-Cymene in the analyte column?. Table 2 has now been included as Supplementary Material (S4) and the layout has been fixed. Table 3, LC-MS/MS data can be eliminated and just discussed as a footnote. EO (% m/V) and SEO (ug/cps) should have the same units. Following the Reviewer’s suggestion, LC-MS/MS data have now been eliminated from Table 3 and now they are discussed in the following paragraph. EO concentrations are now expressed as µg/100 µL in order to improve consistency between units of measure of the concentrations in the analysed samples.
Discussion. Too short, It should be ore comparative with preceding studies on chemical characterization and antibiotic activity (DOI: 10.7324/JABB.2017.50203, 10.1016/j.foodres.2017.03.006, 10.1002/fsn3.1007; PMID: 25870697) and antispasmodic natural extracts, including thyme (http://journal.skums.ac.ir/article-1-610-en.html; http://eprints.skums.ac.ir/934/). This paragraph has been added, as suggested by the Reviewer. References. Certain references are not formatted according to biomolecules. Improve the number of references with less of 10y old up to 80% (currently 32% are ≥10 y old) Reference format has now been accurately checked. Some references dating before 2010 cannot be deleted because are referring to methods. |
Reviewer 2 Report
The authors qualitatively and quantitatively identified the major components in the essential oil and studied its effect in in vivo model. However, some points should be addressed:
- The authors mentioned shortly that the EO was obtained by steam distillation, please elaborate.
- In Table three, please explain what does it mean by (% m/V).
- In Table three, the authors quantified thymol and carvacrol in EO and EO-based formulation using two techniques, and they got almost the same concentration of the two components. However, they were expressed in two different ways (% m/V and µg/cps); please revise.
- The authors quantified thymol and carvacrol only using LC-MS. Yes, they are major, but also 6, p-cymene is one of the main components, and it exerts antibacterial and selective spasmolytic properties. https://onlinelibrary.wiley.com/doi/abs/10.1111/j.1750-3841.2010.02022.x
- Though the study characterized the major components only, thymus oil is rich in several minor compounds, and they might have a significant rule in biological activities, please discuss.
- The excipient contains ascorbic acid that might show activities at high concentrations; please discuss.
Author Response
Bologna, May 25, 2020
Revision of Manuscript ID: biomolecules-798449
“Thymus vulgaris L. essential oil-based formulation: chemical analysis and in vitro intestinal contractility study”
By Matteo Micucci, Michele Protti, Rita Aldini, Maria Frosini, Ivan Corazza, Carla Marzetti, Laura Beatrice Mattioli, Gabriella Tocci, Alberto Chiarini, Laura Mercolini and Roberta Budriesi
Dear Editor,
We have carefully read the comments of the Reviewers and we are grateful for the useful criticisms that allowed us to improve the quality of this study. The manuscript has been revised, and we hope that in the present form it is suitable for publication.
Below we report our answers to specific point raised by the Reviewers.
Thank you again for your attention and we look forward to hearing from you soon.
Yours faithfully,
Roberta Budriesi, Prof.
|
Reviewer 2: |
|
Comments and Suggestions for Authors The authors qualitatively and quantitatively identified the major components in the essential oil and studied its effect in in vivo model. However, some points should be addressed: 1. The authors mentioned shortly that the EO was obtained by steam distillation, please elaborate. 2. In Table three, please explain what does it mean by (% m/V). % m/V has now been changed to µg/100 µL in order to improve consistency between units of measure of the concentrations in the analysed samples. 3. In Table three, the authors quantified thymol and carvacrol in EO and EO-based formulation using two techniques, and they got almost the same concentration of the two components. However, they were expressed in two different ways (% m/V and µg/cps); please revise. As can be seen from Table 3 the results are consistent between the two employed techniques when applied to the same sample (i.e. thymol in EO: 43.3 µg/100 µL with CEC-DAD vs. 43.5 µg/100 µL with LC-MS/MS; carvacrol in EO: 20.7 µg/100 µL with CEC-DAD vs. 21.0 µg/100 µL with LC-MS/MS; thymol in formulation: 210.2 µg/cps with CEC-DAD vs. 213.3 µg/cps with LC-MS/MS; carvacrol in formulation: 99.7 µg/cps with CEC-DAD vs. 102.2 µg/cps with LC-MS/MS). On the other hand, the units of measure are consistent among the concentrations in the same kind of sample. The LC-MS/MS results have now been eliminated from Table 3 and included in the subsequent paragraph, following the suggestion of another Reviewer. 4. The authors quantified thymol and carvacrol only using LC-MS. Yes, they are major, but also 6, p-cymene is one of the main components, and it exerts antibacterial and selective spasmolytic properties. https://onlinelibrary.wiley.com/doi/abs/10.1111/j.1750-3841.2010.02022.x Thymol and carvacrol have been quantified by CEC-DAD together with other 10 minor compounds (including p-cymene), as described throughout the manuscript and as reported in Table 3, while LC-MS/MS analysis was exploited in order to confirm main components thymol and carvacrol concentrations and to prove the effectiveness of CEC-DAD analysis. 5. Though the study characterized the major components only, thymus oil is rich in several minor compounds, and they might have a significant rule in biological activities, please discuss. The Authors believe that, in the context of a targeted quali-quantitative analysis, the selection of 12 analytes belonging to different chemical classes represents a sufficiently large and representative set of bioactive compounds present in thymus EO that can therefore be correlated with the biological activities described in this research work. Following the issue raised by the Reviewer, this concept has now been better clarified in the Introduction section. 6. The excipient contains ascorbic acid that might show activities at high concentrations; please discuss. Ascorbic acid exerts also a direct inhibitory activity towards several bacteria, including those studied in this paper. However, MIC values correspond to concentrations much lower than those used in our study. [Abbas HA, Kadry AA, Shaker GH, Goda RM. Impact of specific inhibitors on metallo-β-carbapenemases detected in Escherichia coli and Klebsiella pneumoniae isolates. Microb Pathog. 2019; 132: 266‐274] In addition, ascorbic acid in present as part of the excipients that have been tested for their antimicrobial effects and produced no inhibition, likely to its very low concentration. |
Round 2
Reviewer 1 Report
Thank you very much for having accepted most suggestions of this reviewer. The following inquiry was not answered nor modified within the article:
Graphs A (EO and SEO) and B (otilonium bromide) from Figures 3 and 5 and graphs A-D in Figure 11 should have the same units, regardless of magnitude differences.
Author Response
Bologna, May 28, 2020
Revision of Manuscript ID: biomolecules-798449
“Thymus vulgaris L. essential oil-based formulation: chemical analysis and in vitro intestinal contractility study”
By Matteo Micucci, Michele Protti, Rita Aldini, Maria Frosini, Ivan Corazza, Carla Marzetti, Laura Beatrice Mattioli, Gabriella Tocci, Alberto Chiarini, Laura Mercolini and Roberta Budriesi
Dear Editor,
We have carefully read the comments of the Reviewers and we are grateful for the useful criticisms that allowed us to improve the quality of this study. The manuscript has been revised, and we hope that in the present form it is suitable for publication.
Below we report our answers to specific point raised by the Reviewers.
Thank you again for your attention and we look forward to hearing from you soon.
Yours faithfully,
Roberta Budriesi, Prof.
|
Reviewer 1: |
|
Comments and Suggestions for Authors Thank you very much for having accepted most suggestions of this reviewer. The following inquiry was not answered nor modified within the article: Graphs A (EO and SEO) and B (otilonium bromide) from Figures 3 and 5 and graphs A-D in Figure 11 should have the same units, regardless of magnitude differences. We really apologize with Referee, but due to a mere forget fulness we omit to amend figures as requested. As OB is active at very low concentration, it was not possible to use the same scale unit on X-axes (i.e. mg/mL), unless using a too wide range of values. We have however amended by using ng/mL (Figure 3, panel B) or µg/mL (Fig 5, panel B), which still allow a correct comparison of the data. For Figure 11, in which the sigmoidal fitting is depicted in panel B and C, it is mandatory to use log[M] of OB in order to correctly calculate the sigmoidal curve. In the same panels, however, we have highlighted the concentration in ng or µg/mL of OB present in the samples, hoping that in this way it is easier to compare the SEO and EO effects vs those of OB to the reader. Finally, all the figures have been embedded in the manuscript at the highest possible resolution. In case this is not sufficient, original files can be provided. |
Reviewer 2 Report
- In the revised manuscript, the authors deleted all information about the extraction method of the essential oil, please elaborate.
- p-Cymene is one of the main components, and it exerts antibacterial and selective spasmolytic, please discuss. properties. https://onlinelibrary.wiley.com/doi/abs/10.1111/j.1750-3841.2010.02022.x
Author Response
Bologna, May 28, 2020
Revision of Manuscript ID: biomolecules-798449
“Thymus vulgaris L. essential oil-based formulation: chemical analysis and in vitro intestinal contractility study”
By Matteo Micucci, Michele Protti, Rita Aldini, Maria Frosini, Ivan Corazza, Carla Marzetti, Laura Beatrice Mattioli, Gabriella Tocci, Alberto Chiarini, Laura Mercolini and Roberta Budriesi
Dear Editor,
We have carefully read the comments of the Reviewers and we are grateful for the useful criticisms that allowed us to improve the quality of this study. The manuscript has been revised, and we hope that in the present form it is suitable for publication.
Below we report our answers to specific point raised by the Reviewers.
Thank you again for your attention and we look forward to hearing from you soon.
Yours faithfully,
Roberta Budriesi, Prof.
|
Reviewer 2: |
|
Comments and Suggestions for Authors 1. In the revised manuscript, the authors deleted all information about the extraction method of the essential oil, please elaborate. We apologize with the referee for the mistake. During revision, we deleted the sentence “The used essential oil has been obtained by steam distillation of the summit flowers.” that has been inserted again. 2. p-Cymene is one of the main components, and it exerts antibacterial and selective spasmolytic, please discuss. properties. https://onlinelibrary.wiley.com/doi/abs/10.1111/j.1750-3841.2010.02022.x We thank the referee for having underlined the importance of p-cymene that was not discussed in the first version. The spasmolytic effect of SEO, may be related, atleast in part, to p-cymene, as this phytochemical is highly present in the studied formulation. This activity has been discussed as require “…..To the SEO spasmolytic action also contributes to monoterpenes for which antispasmodic activity has already been shown [Astudillo, A.; Hong, E.; Bye, R.; Navarrete, A. Antispasmodic activity of extracts and compounds of Acalypha phleoides Cav. Phythoter. Res. 2004, 18, 102-106.]. In particular, in SEO there is an interesting amount of p-cymene (Table 1) to which the literature attributes antispasmodic action through interaction with receptors directly involved in the control of motility such as cholinergic ones [Rivero‐Cruz, I.; Duarte, G.; Navarrete, A.; Bye, R.; Linares, E.; Mata, R. Chemical Composition and Antimicrobial and Spasmolytic Properties of Poliomintha longiflora and Lippia graveolens Essential Oils. J. Food Sci. 2011, 76, C309-C317.]. Also the antimicrobial activity of p-cymene as antimicrobial agent has been described as required “…. and for p-cymene, which has been proven to possess interesting in vitro antimicrobial activity [Rivero‐Cruz, I.; Duarte, G.; Navarrete, A.; Bye, R.; Linares, E.; Mata, R. Chemical Composition and Antimicrobial and Spasmolytic Properties of Poliomintha longiflora and Lippia graveolens Essential Oils. J. Food Sci. 2011, 76, C309-C317.].
|